# Characterization and Exploration of the Flavor Profiles of Green Teas from Different Leaf Maturity Stages of *Camellia sinensis* cv. Fudingdabai Using E-Nose, E-Tongue, and HS-GC-IMS Combined with Machine Learning

**DOI:** 10.3390/foods14162861

**Published:** 2025-08-18

**Authors:** Xiaohui Liu, Mingzheng Huang, Weiyuan Tang, Yucai Li, Lun Li, Jinyi Xie, Xiangdong Li, Fabao Dong, Maosheng Wang

**Affiliations:** College of Food and Pharmaceutical Engineering, Guizhou Institute of Technology, Guiyang 550025, China; liuxiaohui0908@hotmail.com (X.L.); huangmingzheng@git.edu.cn (M.H.); twygycn@126.com (W.T.); liyucai2860@hotmail.com (Y.L.); 13037812076@163.com (L.L.); 15761392354@163.com (J.X.); lxd527@126.com (X.L.)

**Keywords:** green tea, leaf tenderness, volatile compounds, HS-GC-IMS, rOAV, machine learning, Fudingdabai, KEGG pathway, flavor markers, tea grading

## Abstract

Understanding how leaf maturity affects the flavor attributes of green tea is crucial for optimizing harvest timing and processing strategies. This study comprehensively characterized the flavor profiles of Fudingdabai green teas at three distinct leaf maturity stages—single bud (FDQSG), one bud + one leaf (FDMJ1G), and one bud + two leaves (FDTC2G)—using a multimodal approach integrating electronic nose, electronic tongue, HS-GC-IMS, relative odor activity value (rOAV) evaluation, and machine learning algorithms. A total of 85 volatile compounds (VOCs) were identified, of which 41 had rOAV > 1. Notably, 2-methylbutanal, 2-ethyl-3,5-dimethylpyrazine, and linalool exhibited extremely high rOAVs (>1000). FDQSG was enriched with LOX (lipoxygenase)-derived fresh, grassy volatiles such as (Z)-3-hexen-1-ol and nonanal. FDMJ1G showed a pronounced accumulation of floral and fruity compounds, especially linalool (rOAV: 7400), while FDTC2G featured Maillard- and phenylalanine-derived volatiles like benzene acetaldehyde and 2,5-dimethylfuran, contributing to roasted and cocoa-like aromas. KEGG (Kyoto Encyclopedia of Genes and Genomes) analysis revealed significant enrichment in butanoate metabolism and monoterpenoid biosynthesis. Random forest–SHAP analysis identified 20 key flavor markers, mostly VOCs, that effectively discriminated samples by tenderness grade. ROC–AUC validation further confirmed their diagnostic performance (accuracy ≥ 0.8). These findings provide a scientific basis for flavor-driven harvest management and the quality-oriented grading of Fudingdaibai green tea.

## 1. Introduction

Green tea, a globally consumed non-fermented tea product, is prized for its distinctive fresh, floral, and umami sensory attributes [1,2], as well as its significant health benefits, including antioxidant [3,4], anti-inflammatory [5], neuroprotective [6], anticancer [7], cardioprotective [8], antihyperglycemic, and anti-obesity effects [4]. These nutritional and sensory qualities are closely linked to the cultivar and maturity stage of the tea leaves used for processing. Among the diverse tea cultivars in China, *Camellia sinensis* cv. Fudingdabai stands out as a high-quality variety due to its favorable bud and leaf morphology [9,10]. It is widely cultivated not only in its native Fujian Province but also in regions such as Guizhou, where distinct geographical conditions contribute to unique flavor signatures and nutritional profiles of processed teas derived from this cultivar [11,12]. The Guizhou region, characterized by its low latitude, high altitude, and limited sunshine, provides an ideal ecological niche for premium tea production. Under this terroir, Fudingdaibai green tea (FDG) develops distinctive sensory attributes shaped by the synergy between cultivar and processing techniques [12,13,14].

A critical agronomic and quality-related factor influencing green tea is leaf maturity stage, typically categorized as single bud, one bud with one leaf, and one bud with two leaves. The maturation stage of tea shoots directly affects the chemical composition of flavor precursors and secondary metabolites [15,16,17]. Prior studies have demonstrated that teas made from more tender shoots generally exhibit enhanced sweetness and umami, along with lower bitterness, which are associated with the accumulation of metabolites such as maltitol and L-aspartic acid [18]. Similarly, in yellow tea, tender buds have been shown to possess higher antioxidant and α-glucosidase inhibitory activity, while mature leaves exhibit greater levels of Maillard reaction-derived heterocyclic volatiles [19]. Recent research also suggests that the distribution of chiral aroma compounds in green tea varies systematically with leaf maturity [20]. Additionally, Xu et al. [17] found that green teas made from tender shoots had higher amino acid and polyphenol content, yielding fresher and sweeter taste profiles, whereas mature leaf teas exhibited intensified bitterness and astringency. Similar maturity-dependent changes in flavor-active compounds and bioactivity have been documented in *Lithocarpus litseifolius* (sweet tea) [21], mulberry leaf tea [22], black tea [23], and white tea [24], suggesting that leaf maturity is a general determinant of tea flavor and functional properties.

Despite growing interest in the relationship between leaf maturity and tea quality, a systematic flavoromic investigation of how varying leaf maturity stages affect the flavor characteristics, volatile profiles, sensory attributes, and related metabolic pathways in Fudingdabai green tea (FDG)—especially from the Guizhou region—remains lacking. Clarifying these relationships is critical for identifying quality markers and establishing rapid grading frameworks, with important implications for harvest scheduling and processing control.

Recent advancements in flavoromics have enabled more comprehensive tea flavor evaluation. Headspace gas chromatography–ion mobility spectrometry (HS-GC-IMS) has emerged as a powerful analytical platform owing to its high sensitivity, fast analysis time, and minimal sample preparation [2,25,26,27,28]. This technique excels in detecting key volatiles such as aldehydes and ketones, and has been successfully applied to characterize the aroma of green, white, oolong, black, and dark teas, as well as baijiu and meat products [2,25,26,27,28,29,30,31,32]. In parallel, electronic nose (E-nose) and electronic tongue (E-tongue) systems enable rapid quantification of overall aroma and taste features, thereby overcoming the limitations of traditional sensory evaluation [31,32]. Moreover, relative odor activity value (rOAV) analysis provides a quantitative estimation of the contribution of each volatile to the overall aroma profile [25,33]. However, the standalone application of these techniques is often insufficient to fully capture the complexity of food flavor systems.

The integration of machine learning (ML) algorithms presents an effective solution for decoding such multidimensional datasets. Algorithms such as random forest (RF), support vector machine (SVM), k-nearest neighbor (KNN), Gaussian naïve Bayes (GNB), logistic regression (LR), and decision tree (DT) have shown strong performance in modeling nonlinear and high-dimensional flavor data, while also enabling the selection of key discriminative features [34]. These tools have been increasingly used to evaluate and predict flavor quality in baijiu, tea, and other food products [25,26,34,35,36]. Notably, ensemble modeling strategies combining multiple algorithms can further enhance prediction accuracy and reduce model bias [34].

Against this backdrop, the present study aimed to investigate the flavor characteristics of FDG processed from leaves of three different maturity stages—single bud (FDQSG), one bud plus one leaf (FDMJ1G), and one bud plus two leaves (FDTC2G)—harvested from Guizhou Province. By integrating HS-GC-IMS, E-nose, E-tongue, rOAV calculation, and multiple machine learning models, we sought to (1) characterize how tenderness affects the overall flavor profile of FDG, (2) identify key flavor markers associated with leaf maturity, and (3) elucidate potential biosynthetic pathways of differential volatile compounds. The findings offer novel insights into the flavor basis of tea grading and provide a theoretical foundation and technical support for optimized harvest timing and tailored processing of FDG.

## 2. Materials and Methods

### 2.1. Tea Samples and Chemical Reagents

Three green tea samples representing different leaf maturity stages were processed from the fresh leaves of *Camellia sinensis* ‘Fudingdabai’. The harvested fresh shoots included single bud (FDQSG, Fuding Que She), one bud with one leaf (FDMJG, Fuding Maojian), and one bud with two leaves (FDTC2G, Fuding Taicha). All raw tea materials were collected in March and April 2023 from nine-year-old tea plants grown at the Hongfengshanyun Tea Farm in Qingzhen, Guizhou Province, China (106°22′ E, 26°31′ N). To ensure consistency and representativeness, all samples were harvested from equivalent positions on the tea bushes and then thoroughly homogenized prior to analysis. The processed green tea samples were stored at –4 °C until further evaluation. For instrumental analysis, each group was tested in triplicate for the electronic nose (E-nose) and HS-GC-IMS, and in quintuplicate (n = 5) for the electronic tongue (E-tongue) to ensure measurement accuracy and repeatability.

All chemical reagents were of chromatographic or analytical grade. Ethyl decanoate (≥98%, GC grade, TCI, Shanghai, China) was used as the internal standard for volatile compound analysis. Standard n-ketones for retention index (RI) calibration—2-butanone (99.5%), 2-pentanone (99.5%), 2-hexanone (99.5%), 2-heptanone (98%), 2-octanone (99%), and 2-nonanone (99%)—were purchased from Sinopharm Chemical Reagent Co., Ltd. (Shanghai, China). Sodium chloride (NaCl) and absolute ethanol were obtained from Jinshan Chemical Co., Ltd. (Chengdu, China), and distilled water was provided by Wahaha Group Co., Ltd. (Hangzhou, China).

### 2.2. Electronic Tongue Sensory Analysis

Taste attributes of green tea infusions were assessed using a taste sensing system (SA402B Electronic Tongue, INSENT Inc., Kanagawa, Japan), which is equipped with five test sensors (AAE, CTO, CAO, COO, AE1) and two reference electrodes. These sensors detect the intensity of umami, saltiness, sourness, bitterness, and astringency, as well as aftertaste characteristics such as aftertaste-bitterness (aftertaste-B), aftertaste-astringency (aftertaste-A), and richness.

Each tea sample (1.5 g) was extracted with 75 mL of boiling distilled water for 5 min. The resulting infusion was filtered through gauze and cooled to room temperature prior to measurement. Five independent replicates were prepared and analyzed for each tenderness type. The operating conditions and measurement procedure of the electronic tongue are detailed in Appendix A.

### 2.3. Electronic Nose Sensory Analysis

The aroma profiles of green tea samples with varying leaf maturity stages were analyzed using a PEN3 electronic nose system (Airsense Analytics GmbH, Schwerin, Germany), which is equipped with a sensor array of ten metal oxide gas sensors. These sensors are sensitive to different classes of volatile compounds, including aromatic benzenes, nitrogen oxides, amines, hydrogen, short/long-chain alkanes, sulfur organics, alcohols, aldehydes, and ketones (Appendix A).

For sample preparation, 0.5 g of liquid nitrogen-ground tea powder was accurately weighed into a 20 mL headspace vial. After adding 5 mL of boiling distilled water, the vial was immediately sealed to allow headspace extraction. The sealed vials were equilibrated in a water bath at 65 °C for 50 min before analysis.

The PEN3 system was operated under the following conditions: sampling interval of 1 s, pre-sampling time of 5 s, measurement time of 150 s, and flushing time of 60 s. Zero-point calibration was performed automatically with clean air for 10 s before each measurement. Both the chamber and injection flow rates were set at 400 mL/min. The measurement lasted 120 s. The chamber was purged with filtered air between runs to restore baseline sensor responses.

### 2.4. HS-GC-IMS Analytical Method

The analysis of volatile organic compounds (VOCs) in green tea samples was conducted using a FlavourSpec^®^ GC-IMS instrument (G.A.S., Dortmund, Germany), equipped with an MXT-5 capillary column (15 m × 0.53 mm × 1.0 μm, Restek, Bellefonte, PA, USA), a headspace autosampler (CTC-PAL3, CTC Analytics AG, Zwingen, Switzerland), and an ion mobility spectrometry detector. For sample preparation, 1.0 g of finely ground tea sample was accurately weighed into a 20 mL headspace vial, to which 20 μL of ethyl decanoate (100 ppm) was added as the internal standard. The vial was incubated at 80 °C for 15 min with agitation at 500 rpm. A 500 μL aliquot of the headspace was injected into the GC-IMS system using a heated syringe (85 °C) in splitless mode.

The GC-IMS operating parameters were as follows: the column temperature was maintained at 60 °C; the IMS drift tube was held at 45 °C; the carrier gas (nitrogen, ≥99.999%) flow rate was programmed as 2 mL/min (0–2 min), increasing linearly to 100 mL/min over 18 min. The drift gas (nitrogen, ≥99.999%) was supplied at a constant flow rate of 150 mL/min. The total analysis time was 20 min. Retention indices (RI) were calculated using a homologous series of n-ketones (C4–C9) under identical chromatographic conditions. All measurements were performed in triplicate to ensure accuracy and reproducibility.

### 2.5. Qualitative and Quantitative Analysis

The identification of volatile compounds was performed using VOCal software (version 0.4.07, rev.# 346; G.A.S., Dortmund, Germany), integrated with both the National Institute of Standards and Technology (NIST 17) mass spectral database and the proprietary GC-IMS library. Retention indices (RI) were calculated by analyzing a homologous series of n-alkanes (C4–C9) under identical chromatographic conditions as the tea samples. VOCs were identified by comparing both their RI values and drift times (Dt) with those in the reference databases, ensuring accurate compound assignment.

For semi-quantitative analysis, the internal standard method was applied, using ethyl decanoate (100 ppm) as the reference. The concentrations of individual VOCs were expressed as relative values based on the ratio of the peak area of each compound to that of the internal standard.

Additionally, rOAVs were calculated to evaluate the sensory relevance of each volatile compound. The rOAV was defined as the ratio of the compound’s relative concentration to its reported odor threshold in water. By integrating both chemical concentration and sensory detection threshold, rOAV provides a more reliable indication of human olfactory perception than concentration alone. In food, fragrance, and environmental odor analysis, this metric enables the prioritization of high-rOAV odorants (even at low concentrations) over abundant but sensorially negligible compounds with low rOAVs. VOCs with rOAV values equal to or greater than 1 were considered to contribute significantly to the overall aroma profile of the green tea samples [25]. To assess the contribution of individual VOCs to the aroma profiles of FDG with different maturity stages, those meeting this criterion were extracted and summarized in Table 1 for further discussion.

### 2.6. Feature Selection and Extraction of Key Flavor Attributes in Green Teas of Different Leaf Maturity Stages

To identify key sensory and chemical markers responsible for the quality differentiation among green tea samples of varying leaf maturity stages, multivariate statistical analysis and machine learning algorithms were employed for feature selection and extraction from datasets obtained through electronic nose, electronic tongue, and HS-GC-MS analyses.

In the domain of machine learning, feature engineering methods are generally classified into two categories: feature extraction and feature selection. Principal component analysis (PCA) and partial least squares discriminant analysis (PLS-DA) were applied as typical feature extraction techniques to reduce data dimensionality while preserving variance and class separation. For feature selection, three methodological frameworks were considered: filter methods, wrapper methods, and embedded methods. Among these, the random forest (RF)-based wrapper method was used to rank the importance of individual features according to their contribution to classification performance.

The most informative features identified were further subjected to classification modeling using six commonly adopted machine learning algorithms: k-nearest neighbors (KNN), random forest (RF), support vector machine (SVM), Gaussian naïve Bayes (GNB), logistic regression (LR), and decision tree (DT). Model performance was evaluated based on receiver operating characteristic (ROC) curves, accuracy, sensitivity, and specificity [34,35,36].

Prior to classification modeling, three complementary feature selection strategies were employed to identify key variables. First, the entire dataset was utilized without prior selection to assess the predictive performance of all available features. Second, feature importance was evaluated using Shapley Additive Explanations (SHAP) derived from the random forest model, and the top 20 features with the highest SHAP values were selected to construct reduced models. Third, each of the top 20 SHAP-ranked features was individually used to train classification models, allowing for the assessment of the independent predictive power of each variable [34,36].

To optimize model performance, hyperparameter tuning was conducted for all classification algorithms using K-fold cross-validation, with the area under the receiver operating characteristic curve (AUC) serving as the primary evaluation metric. Features associated with AUC values between 0.80 and 0.90 were considered moderately discriminative, while those with AUC values above 0.90 were deemed to exhibit strong predictive ability [25].

### 2.7. Data Analysis

Volatile compounds detected by the electronic nose and HS-GC-IMS were analyzed in triplicate, while taste attributes measured using the electronic tongue were conducted in five replicates. All results were expressed as mean ± standard deviation (SD). Statistical significance was assessed using one-way analysis of variance (ANOVA) via SPSS Statistics 26.0 (IBM, Armonk, NY, USA), with a threshold of *p* < 0.05 considered significant.

To visualize GC-IMS results, two-dimensional top-view plots, three-dimensional topographic plots, and differential spectral fingerprints were generated using Reporter and Gallery Plot plug-ins (G.A.S., Dortmund, Germany). These visualizations facilitated comparison of volatile profiles across green tea samples of different leaf maturity stages.

Multivariate statistical analyses were performed using MetaboAnalyst 6.0 [37]. Principal component analysis (PCA) and partial least squares discriminant analysis (PLS-DA) were applied to examine overall clustering and sample discrimination. Additional data visualization methods, including hierarchical cluster analysis (HCA), heatmaps, and volcano plots, were also employed within the MetaboAnalyst platform to identify and interpret significantly different variables. Pie charts and chord diagrams were created using OriginPro 2021 (OriginLab Corporation, Northampton, MA, USA) for the graphical presentation of selected variables and compound associations. K-means clustering and supervised machine learning analyses were conducted using the Biodeep Cloud Platform provided by Panomix (Suzhou, China), enabling feature discrimination and predictive model construction across green tea samples.

## 3. Results and Discussion

### 3.1. Electronic Tongue Analysis

Taste profiles across green tea samples of differing tenderness were characterized using an electronic tongue (Figure 1). Principal component analysis (PCA) revealed that PC1 and PC2 explained 85.5% and 13.9% of variance, respectively, cumulatively capturing 99.6% of the sensory data (Figure 1A). This demonstrates that the first two principal components effectively capture the majority of taste-related variation among samples. Clear separation in the PCA score plot indicates that leaf tenderness significantly influences flavor attributes. Specifically, the single-bud sample (FDQSG) was strongly associated with “umami” characteristics (Figure 1B), likely reflecting enriched amino acid content [38]. The one-bud, one-leaf tea (FDMJ1G) was marked by heightened sourness, astringency, bitterness, and pronounced bitter aftertaste (aftertaste-B), potentially attributable to increased caffeine levels in moderately mature leaves (Figure 1B,C). By contrast, the one-bud, two-leaf tea (FDTC2G) correlated with saltiness, richness, and aftertaste astringency (Aftertaste-A), which may correspond to increased levels of tea polyphenols and catechins in fully developed leaves [39]. These results imply that a comprehensive investigation into the distribution of non-volatile compounds is necessary to elucidate the correlation between taste profiles and chemical constituents in green teas of different tenderness grades.

### 3.2. Electronic Nose Analysis

A supervised PLS-DA model was constructed to classify the aroma profiles of the different samples. The model’s first two latent variables accounted for 78.2% and 20.0% of variance (totaling 98.2%) and clearly separated tenderness groups in the score plot (Figure 2A), indicating strong discriminative capacity. Model robustness was confirmed by cross-validated Q^2^ = 0.988 and R^2^ = 0.992, with 100% classification accuracy; permutation testing (n = 2000) yielded *p* < 5 × 10^−4^, confirming statistical significance (Figure 2C,D).

The bi-plot (Figure 2B,F) revealed clear sensor–odor relationships: FDQSG correlated with sensors W2W, W5S, and W1W; FDTC2G with W5C, W1C, W1S, and W2S; and FDMJ1G overlapped with both but displayed stronger sensitivity to W3C. VIP analysis (VIP > 1) identified W3C (aromatic amines), W3S (long-chain alkanes), W1S (short-chain alkanes), and W6S (hydrogen-related compounds) as key discriminators (Figure 2E, Appendix A).

### 3.3. HS-GC-IMS Analysis

#### 3.3.1. Qualitative and Quantitative VOC Profiling

The overall volatile compound profiles of green teas with different leaf maturity stages were systematically analyzed using HS-GC-IMS. Compound identification was based on retention time, drift time, and retention index values derived from IMS data. Using the Reporter plug-in, both three-dimensional (Figure 3A) and two-dimensional (Figure 3B) topographic visualizations were generated to comprehensively characterize the volatile fingerprints of the tea samples. In the 3D representation, the *X*-axis corresponds to ion drift time, the *Y*-axis to gas chromatographic retention time, and the *Z*-axis to the signal intensity of each volatile compound. The 2D topographic plot offers a more intuitive overview of VOC distribution across the samples, effectively illustrating characteristic fingerprints for green teas of varying leaf tenderness. As shown in Figure 3B, the spectra were normalized and aligned using the reactive ion peak (RIP), which appears at a drift time of 1.0 (marked by a vertical reference line). Each feature point to the right of the RIP represents an individual volatile component detected in the sample. The color gradient from light blue to deep red indicates increasing compound concentration, with deeper red denoting higher relative abundance.

Figure 3 demonstrates the presence of multiple characteristic peaks, indicating that green tea samples of different maturity levels contain a diverse array of volatile organic compounds (VOCs). To further elucidate differences in volatile profiles among the teas, a differential comparison method was employed (Figure 3C), with the single-bud sample (FDQSG) set as the reference. Relative differences in VOC intensity between FDQSG and the other two samples (FDMJ1G and FDTC2G) were computed to generate visual differential plots. In these maps, white regions indicate no significant difference; red regions reflect relative enrichment of specific compounds in the compared sample, whereas blue regions indicate reduced abundance relative to the reference. The differential plots revealed a substantially greater number of blue regions in both FDMJ1G and FDTC2G compared with FDQSG, suggesting that increased leaf maturity is associated with a general decline in the concentration of many volatile aroma compounds. Further identification and visualization of these decreasing compounds require detailed examination of the VOC fingerprint maps.

In total, 85 VOCs were detected (Appendix A), comprising 8 terpenes, 19 aldehydes, 13 ketones, 9 alcohols, 10 esters, 6 furans, 3 aromatics, 3 sulfur compounds, 3 pyrazines, 2 acids, 1 heterocyclic (2-acetyl-1-pyrroline), and 8 unidentified peaks. Notably, five VOCs—nonanal-D, heptanal-D, octanal-D, n-hexanol-D, and pentan-1-ol-D—were identified as forming dimers, a phenomenon attributed to proton-bound dimerization (2M+H)^+^ at elevated concentrations [27]. The dominant compound classes were aldehydes (22.4%), ketones (15.3%), alcohols (11.8%), and esters (11.8%), with terpenes at 8.2% (Figure 4A).

Quantitative analysis revealed the highest total VOC content in FDMJ1G (22.03 mg/L), followed by FDQSG and FDTC2G. FDQSG was notable for its aldehyde, sulfur compound, pyrazine, and acid content. FDMJ1G was enriched in terpenes, ketones, furans, aromatics, and unknown compounds, whereas FDTC2G had higher levels of alcohols, esters, heterocyclics (e.g., 2-acetyl-1-pyrroline), and sulfides—including compounds associated with “popcorn”, “meaty”, and “coffee” aromas (Appendix A, Figure 4B). These compositional differences suggest the involvement of distinct metabolic pathways across different leaf maturity levels, which may have important implications for optimizing processing strategies and guiding product classification and grading in green tea production.

To further identify characteristic volatile compounds and distinguish aroma-active substances among green tea samples with different leaf maturity stages, fingerprint profiles were constructed to visualize the concentration changes of individual VOCs (Figure 5). Specifically, regions I, III, and IV correspond to the potential characteristic VOCs of FDQSG, FDMJ11G, and FDTC2G, respectively. Notably, region I, representing FDQSG (single bud), featured the highest number of discriminative compounds (21 in total), predominantly composed of C5–C9 short-chain aldehydes and ketones that impart fresh and green notes. Additionally, short-chain esters (e.g., butyl propanoate) and unsaturated aldehydes (e.g., (E)-2-octenal) contributed fruity and ester-like aromas, while unsaturated alcohols such as oct-1-en-3-ol provided mushroom, earthy, and green sensory characteristics. These compounds are recognized as key odor-active constituents in Nen Xiang (NX), Li Xiang (LX), and Qing Xiang (QX) style green teas [40]. Furthermore, pyrazines (e.g., 2-ethyl-6-methylpyrazine) and furans (e.g., 2-n-butylfuran), common products of Maillard reactions, contributed roasty and nutty aromas [41]. Collectively, these volatiles define the fresh and green aroma characteristics typical of single-bud green teas.

In contrast, Region III, associated with FDMJ11G (one bud with one leaf), contained 12 representative VOCs, including methyl salicylate, linalool, 2-butoxyethanol, n-propyl acetate, 2-methyl-2-pentenal, beta-pinene, 2-acetylfuran, furfural, 3-methylthiopropanal, and three unknown compounds. As an intermediate maturity stage, FDMJ11G exhibited a volatile profile suggestive of a metabolic transition between young and mature leaves. This was characterized by terpenoid dominance (e.g., linalool, beta-pinene) and a balance between esters and aldehydes, reflecting a dynamic conversion from alcohols to aldehydes and subsequently to esters. Linalool and methyl salicylate contributed complex floral attributes (citrus, floral, rose, wintergreen, minty), which distinguished this profile from the fresh-green aroma of FDQSG. Additionally, furfural and 2-acetylfuran provided roasted undertones, mitigating excessive grassy notes. Intriguingly, Chen-Yang Shao’s study [20] on baked green teas with varying tenderness reported a significant increase in methyl salicylate with leaf maturity, which differs from our findings—likely due to varietal differences.

Region IV represented the characteristic VOCs of FDTC2G (one bud with two leaves), which included 3-pentanol, 2-acetyl-1-pyrroline, 2,5-dimethylfuran, benzene acetaldehyde, 2-butanone, and ethylsulfide. The VOC profile of this group, derived from more mature leaves, reflected pronounced Maillard reaction activity during processing, facilitating the formation of compounds such as 2-acetyl-1-pyrroline (popcorn-like) and 2,5-dimethylfuran meaty, gravy, roasted beef juice). The accumulation of 3-pentanol and 2-butanone suggested elevated lipid oxidation in FDTC2G compared to the less mature counterparts. 3-Pentanol (herbal) and benzene acetaldehyde (green, floral, sweet) co-occurred with 2-acetyl-1-pyrroline (meaty, roasted) to create a layered and mellow flavor profile for two-leaf green tea. Previous research has identified dimethyl sulfide (cooked corn-like) as a key contributor to the fresh and delicate aroma of green tea [28,42], while ethylsulfide—another key VOC in FDTC2G—conveyed coffee- and meat-like notes. The conversion pathway between dimethyl sulfide and ethylsulfide may be critical for modulating the aroma type and quality of FDG, warranting further investigation.

Furthermore, Regions II and V reflected common VOCs shared by all three green tea types. Based on their concentrations and aroma characteristics, six major shared compounds were identified with levels exceeding 1000 µg/L: linalool (citrus, floral, rose), heptanal-D (green), 2-methylbutanal (cocoa), 2-propanone (ethereal, apple, pear), mesityl oxide (vegetal), and methyl acetate (ethereal sweet, fruity). Among these, linalool and heptanal are well-established as key aroma-active compounds in green tea [2,43,44]. Additionally, 2-methylbutanal (VIP = 2.70) and 3-methylbutanal (VIP = 1.30) have been reported as differential markers between yellow teas of different tenderness (LYT vs. BYT/SYT) [19], further suggesting that leaf maturity strongly influences the accumulation of 2-methylbutanal.

Additionally, region IV contained compounds commonly expressed in both FDMJ11G and FDTC2G, notably benzaldehyde (sharp sweet, almond, cherry) and beta-ocimene (floral). Benzaldehyde has been cited as a marker for chestnut-like aroma in green tea [45], while beta-ocimene, with OAV > 1, is a key floral contributor in flower-scented green teas [40].

Overall, fingerprint-based VOC profiling provided valuable insights into the compositional and sensory differentiation among green teas with varying tenderness. However, it should be noted that aroma perception is influenced not only by compound concentration but also by the odor threshold in a given matrix. Thus, high-concentration compounds may not necessarily dominate aroma perception, and low-concentration compounds could still play key roles. Quantitative verification of these contributions requires further rOAV (relative Odor Activity Value) analysis.

#### 3.3.2. Analysis of VOCs Variation Among FDG of Different Tenderness

Hierarchical cluster analysis (HCA) of all detected VOCs effectively distinguished the green tea samples by tenderness level (Figure 6A). Notably, the two more mature samples, FDMJ1G and FDTC2G, clustered together on the same branch, while the most tender sample, FDQSG, formed a separate clade, indicating clear differentiation based on VOC composition. Subsequent K-means clustering divided the complete set of VOCs into four distinct clusters (Figure 6B), each exhibiting unique trends associated with leaf tenderness. Specifically, Cluster 2 contained VOCs whose concentrations increased as leaf tenderness decreased; Cluster 4 showed the opposite trend, with a decline in compound abundance as tenderness decreased. Meanwhile, Cluster 1 exhibited a U-shaped pattern (initial decline followed by an increase), and Cluster 3 demonstrated an inverted U-shape (initial increase followed by a decrease).

Focusing on Cluster 2, which included 12 compounds that increased with decreasing tenderness (see Appendix A), this group exhibited a wide spectrum of sensory attributes such as citrus, cocoa, fruity, roasted, coffee, and popcorn-like aromas. Several compounds also possessed functional properties beyond aroma: for example, Octanal-M is known for its antioxidant and antimicrobial activity [46], while 3-Pentanol has been reported to stimulate plant immune responses [47]. In contrast, Cluster 4, comprising 37 VOCs that decreased in abundance with increasing leaf maturity, was characterized predominantly by fresh, green, fatty, fruity, and earthy notes. This suggests a substantial loss of freshness and green-earthy qualities as tenderness diminishes.

Specifically, Cluster 1, with 11 VOCs—primarily aldehydes, alcohols, and ketones—displayed a pattern of decreasing and then increasing concentrations across the tenderness gradient. These compounds were associated with floral, green, and citrusy aromas. Whereas, Cluster 3, encompassing 25 VOCs, exhibited greater structural diversity and included terpenes, short-chain aldehydes/ketones, and aromatic compounds. This cluster was dominated by floral-terpenic, woody, and bready notes, with key compounds such as linalool (citrus, floral, woody), cis-linalool oxide (earthy, floral, woody), β-ocimene (citrus, tropical, green), and β-pinene (pine, woody, resinous). Preliminary analysis suggested that the FDMJ1G sample exhibited notably higher expression levels of terpene-related compounds. These dynamic clustering patterns offer valuable insights into the transformation of volatile profiles across different maturity levels of green tea, highlighting both flavor transitions and potential bio-functional implications.

To further elucidate the intergroup differences in VOCs among green teas of varying leaf tenderness, volcano plots were constructed based on fold change (FC) and statistical significance (adjusted *p* < 0.05, |log_2_FC| ≥ 1). A total of 17 VOCs showed significant differences between FDMJ1G and FDQSG, of which 8 were upregulated and 9 were downregulated (Figure 7A). The upregulated compounds primarily included terpenes such as linalool and β-pinene, which are associated with enhanced plant defense and antimicrobial activity [48,49], contributing floral and dry woody notes. Aromatic aldehydes (e.g., benzaldehyde) and furans (e.g., furfural) impart bitter almond, bready, and baked aromas, while esters (e.g., ethyl acetate, n-propyl acetate) and ketones (e.g., 2-butanone) enrich fruity characteristics. In contrast, the downregulated VOCs were predominantly green leaf volatiles responsible for fresh-green notes, including (E)-2-pentenal, (E)-hept-2-enal, 4,5-dihydro-3(2H)-thiophenone, and several fruity, earthy, herbal, and spicy compounds such as 2-methylbutanol acetate, butyl propanoate, 2-octanone, and 1-penten-3-one.

Similarly, in the FDTC2G vs. FDQSG comparison (Figure 7B), 10 VOCs were significantly upregulated and 10 were downregulated. The upregulated compounds—linalool, methyl isobutyl ketone, benzaldehyde, 2,5-dimethylfuran, 2-butanone, ethyl acetate, 3-pentanol, 2-acetyl-1-pyrroline, ethylsulfide, and benzene acetaldehyde—collectively contributed floral, almond, bready, sweet fruity, toasted grain, coffee, meaty, and honey-like aromas. On the other hand, downregulated compounds included 2-methylbutanol acetate, 1-penten-3-one, 2-octanone, (E)-2-pentenal, hexanal, 4,5-dihydro-3(2H)-thiophenone, butyl propanoate, 2-ethyl-6-methylpyrazine, and (E)-hept-2-enal, which are associated with green, herbal, fatty, and roasted potato–like aromas. The reduction of these compounds indicates a decline in the fresh, fruity, and green sensory qualities.

By contrast, relatively fewer differential compounds were observed between the two more mature samples (FDTC2G vs. FDMJ1G). Only four VOCs showed significant upregulation in FDTC2G (2,5-dimethylfuran, 3-pentanol, 2-acetyl-1-pyrroline, and benzene acetaldehyde), while five VOCs (including (E)-2-pentenal, furfural, n-propyl acetate, and 2-methyl-2-pentenal) were significantly downregulated (Figure 7C). The upregulated compounds contributed roasted, nutty, and fermented notes, indicating a metabolomic shift characteristic of more mature tea leaves. The downregulated compounds were associated with fresh and fruity aromas, echoing the trend observed between FDQSG and the more mature samples.

In summary, comprehensive analysis revealed distinct volatile compound signatures across the three tenderness grades of green tea, demonstrating clear metabolic stratification associated with leaf maturity. In greater detail, the volatile profile of FDQSG (single bud) is predominantly characterized by high levels of C6 aldehydes and alcohols (e.g., hexanal, (Z)-3-hexenol) and monoterpenes (e.g., nerol), which jointly contribute to the fresh, green, fruity, and floral notes typically associated with cut grass and citrus aromas. By contrast, FDMJ1G (one bud plus one leaf) represents an intermediate stage of leaf maturity, where the aroma profile reflects a transition in metabolic activity. The relative balance between terpenes and esters, along with the emergence of Maillard reaction–derived compounds such as benzaldehyde and furfural, results in a more complex and enriched aroma bouquet. Notably, compounds like linalool (floral), β-pinene (woody), and benzaldehyde (almond-like) are markedly elevated, while the content of C6 aldehydes shows a progressive decline. In the case of FDTC2G (one bud plus two leaves), the volatile composition is dominated by Maillard reaction products (e.g., 2,5-dimethylfuran, 2-acetyl-1-pyrroline), fermentation-related esters (e.g., ethyl acetate), and microbial metabolites (e.g., 3-pentanol), collectively imparting roasted, nutty, and savory characteristics. Interestingly, while 2-ethyl-6-methylpyrazine—a key compound contributing to roasted potato–like aromas—is known to enhance roasted notes, its concentration decreases with increasing leaf maturity, implying the involvement of a more intricate regulatory mechanism that warrants further investigation.

Overall, the findings highlight that FDQSG is particularly rich in fresh, green, and fruity aromatic attributes, making it highly suitable for low-temperature fixation processes that aim to retain delicate floral volatiles. This observation is consistent with previous reports on large-leaf black teas with varying degrees of tenderness [18], although the specific compounds contributing to freshness and fruitiness may vary depending on cultivar and processing techniques. FDMJ1G exhibits a more balanced and multidimensional aroma profile, integrating both floral and fruity characteristics, indicative of an intermediate metabolic state. In contrast, FDTC2G is enriched in nutty, roasted, and fermented sweet–meaty notes, reflecting greater metabolic transformation during processing. These findings are also in agreement with previous research on yellow teas processed from leaves of different maturity levels [19], where teas made from more mature leaves were associated with enhanced roasted aromas, while bud-only teas demonstrated higher intensities of fresh and floral sensory attributes.

These results suggest that leaf maturity markedly influences volatile metabolite composition and aroma profile. Notably, some VOCs with lower concentrations may still significantly impact scent if their odor activity (rOAV) is high, warranting further sensory threshold validation.

#### 3.3.3. Multivariate Statistical Analysis

To further investigate the differential VOCs and their evolution across green teas of varying tenderness, a partial least squares discriminant analysis (PLS-DA) model based on VOCs was employed (Figure 8). The PLS-DA model yielded high explanatory and predictive power, with R^2^X = 0.909, R^2^Y = 0.999, and Q^2^ = 0.998. Additionally, a 200-time permutation test was conducted to assess model robustness, showing a Q^2^ regression intercept of −0.57 on the *Y*-axis, which is below zero (Figure 8B), indicating that the model is reliable and not overfitted. As shown in the score plot (Figure 8A), PC1 accounted for 61.6% of the variance, while PC2 contributed 29.3%, together explaining 90.9% of the total variance. This demonstrates that the first two principal components sufficiently represent the original dataset. The three green tea samples—FDQSG, FDMJ1G, and FDTC2G—were well-separated in the model without any overlap, indicating that the HS-GC-IMS method for VOC acquisition and detection provided strong classification ability for tea samples of different leaf maturity stages. Moreover, the strong clustering of VOC profiles within each group reflects high repeatability and stability in VOC detection for samples of the same tenderness level.

The loading plot (Figure 8C) illustrates the contribution of highly correlated VOCs to the principal components, further clarifying the differences in volatile profiles among the three green tea samples of varying tenderness. The plot highlights the VOCs most strongly associated with each sample group—FDQSG, FDMJ1G, and FDTC2G—offering additional insight into their respective aroma characteristics. As shown in Figure 8C, FDQSG exhibits a larger number of highly correlated VOCs compared to the other two groups. These findings are consistent with the characteristic compounds identified in Figure 5. Specifically, FDQSG is predominantly associated with C5–C9 short-chain aldehydes and ketones, short-chain esters, which contribute to its fresh and green aroma. FDMJ1G features a balanced profile of esters and aldehydes, reflecting an intermediate metabolic state. In contrast, FDTC2G is characterized by Maillard reaction products and 3-pentanol, indicating a shift toward roasted, nutty, and fermented flavor attributes as leaf maturity increases.

To further identify key VOCs responsible for differentiating among green tea samples, variable importance in projection (VIP) scores were calculated. VIP scores reflect the contribution and explanatory power of each variable to the classification and discrimination of the sample groups. Compounds with VIP scores > 1 were considered significant discriminators. As shown in Appendix A, 31 VOCs with VIP > 1 were identified as key differential volatile markers in the PLS-DA model based on HS-GC-IMS. These VOCs serve as candidate markers for distinguishing green teas of different leaf maturity stages. These VOCs serve not only as markers for distinguishing green teas of different leaf maturity stages but also as foundational features for building subsequent machine learning models aimed at identifying key characteristic parameters.

#### 3.3.4. rOAV Analysis

To assess the contribution of individual volatile compounds (VOCs) to the overall aroma profile of FDG with different tenderness grades, the relative odor activity value (rOAV) of each compound was calculated (Appendix A). Compounds with rOAVs greater than 1 are generally considered to have a perceptible impact on aroma [33], and thus VOCs meeting this criterion were extracted and summarized in Table 1 for further discussion.

In total, 41 volatile compounds exhibited rOAV values exceeding 1. Among them, FDQSG contained 33 such compounds, FDMJ1G had 32, and FDTC2G had 31. Notably, 30 VOCs with OAV > 1 were common across all three groups, indicating that the majority of the identified volatiles contribute meaningfully to the aroma profile of FDG. Among these, compounds with extremely high rOAV (rOAV > 1000) were primarily aldehydes (e.g., 2-methylbutanal), terpenes (linalool), and pyrazines (2-ethyl-3,5-dimethylpyrazine), which serve as key odorants. Specifically, 2-methylbutanal (chocolate/cocoa-like) and 3-methylbutanal (chocolate, peach) demonstrated remarkably high rOAVs (>1000 and 580–730, respectively), confirming their potent sensory effects. Linalool, a floral terpene, exhibited the highest rOAV across all samples (ranging from 3800 to 7400), despite its relatively low concentration due to its ultra-low odor threshold (OT = 0.22 μg/L), making it a major aroma-contributing compound. Likewise, 2-ethyl-3,5-dimethylpyrazine, with its nutty character and extremely low OT (0.040 μg/L), also showed high rOAVs (1350–1440), indicating its persistent and potent contribution.

Interestingly, linalool peaked in the intermediate maturity sample FDMJ1G, suggesting a metabolic surge at this developmental stage, whereas 2-methylbutanal increased progressively with maturity and reached its maximum in the most mature sample FDTC2G. Pyrazines, in contrast, remained relatively stable across all tenderness stages, indicating their sustained contribution to nutty and roasted attributes.

Furthermore, a clear maturity-dependent shift in dominant aroma-active compounds was observed across the three leaf maturity stages of green tea. In early maturity (FDQSG, single bud), the aroma profile was primarily characterized by fresh, green, floral, and citrus-like attributes. This was largely driven by high rOAVs of terpenes such as linalool (3800) and trans-linalool oxide (4.2), along with C6 aldehydes and alcohols, including hexanal (170) and (Z)-3-hexen-1-ol (67), which contributed grassy and leafy freshness. In addition, the presence of 2-methylbutanal (1100), a cocoa-like aldehyde, imparted a subtle sweet and nutty nuance. These volatiles collectively defined the characteristic fresh and green sensory quality typical of bud-only green teas.

At the intermediate maturity stage (FDMJ1G, one bud plus one leaf), the aroma complexity increased significantly, reflecting a balance between floral, fruity, and nutty notes. Notably, linalool reached its highest rOAV (7400), indicating a floral surge likely linked to transient metabolic activation. Meanwhile, 2-ethyl-3,5-dimethylpyrazine (1400) provided a stable nutty backbone. A substantial decline in hexanal (from 170 to 71; −58% vs. FDQSG) marked the attenuation of green freshness. Concurrently, elevated levels of Maillard- and fermentation-associated compounds such as 2-acetyl-1-pyrroline and ester volatiles (e.g., 2-methylbutanol acetate) contributed toasted, sweet, and fruity nuances, indicating active secondary metabolic transitions.

In contrast, late maturity (FDTC2G, one bud plus two leaves) was marked by a pronounced shift toward roasted, nutty, and savory sensory attributes. The sharp increase in Maillard-derived 2-acetyl-1-pyrroline (from 66 to 160; +142% vs. FDMJ1G) contributed intense popcorn-like notes. The persistent high levels of 2-ethyl-3,5-dimethylpyrazine (~1350) ensured continuity in nutty perception. These changes collectively reflect a maturity-dependent transformation in aroma composition, likely governed by both enzymatic and non-enzymatic reactions during leaf development and subsequent processing.

To further validate the contributions of aroma-active compounds (rOAV > 1) listed in Table 1 to the overall flavor perception of green tea infusions, Pearson correlation analysis was performed between these volatiles and the responses of 10 E-nose sensors (W1C, W3C, W5C, W1S, W2S, W3S, W5S, W6S, W1W, W2W), as well as 8 E-tongue taste attributes (sourness, bitterness, astringency, aftertaste-A, aftertaste-B, umami, richness, saltiness) (Appendix A). Correlation pairs meeting the criteria of |r| ≥ 0.90 and *p* < 0.05 were considered statistically significant.

Notably, 2-acetyl-1-pyrroline showed extremely strong positive correlations with W1S, W2S, and W5C sensors, establishing it as a core marker for “sweet/popcorn-like” aroma recognition by the electronic nose. Simultaneously, it significantly suppressed bitterness, aftertaste-B, and sourness responses, suggesting its synergistic role in reducing bitterness and sourness. Likewise, methyl acetate exhibited positive correlations with richness and saltiness, and a negative correlation with aftertaste-B, indicating its role in reinforcing umami and mellow taste characteristics. In addition, C6–C9 aldehydes (e.g., hexanal, (E)-2-hexenal, pentanal) consistently showed strong positive correlations with the W5S channel, enabling rapid discrimination of tenderness grades. Moreover, ethyl sulfide demonstrated robust positive correlations with multiple sensors (W1S/W2S/W5C/W1C), confirming its distinct “coffee-like” characteristics in E-nose detection. In the taste dimension, ethyl sulfide exhibited strong negative correlations with aftertaste-B and bitterness, indicating that beyond contributing to characteristic coffee-like notes, this compound may also play a modulatory role in suppressing bitterness and lingering astringency, thereby enhancing the overall smoothness and roundness of the tea infusion.

In summary, aroma-active VOCs such as 2-acetyl-1-pyrroline, methyl acetate, C6–C9 aldehydes, and acetate esters exhibited highly significant correlations with key E-nose channels (e.g., W5S, W2W, W1S) and E-tongue taste indices (e.g., richness and aftertaste-B). These strong chemical–sensor couplings highlight their potential as core diagnostic markers for the rapid classification of green tea leaf maturity levels. Previous studies on food flavor have indicated that volatile aroma compounds can modulate taste perception through cross-modal interactions [50,51]. The specific mechanisms by which these volatiles influence taste attributes—such as bitterness, umami, and aftertaste—merit further in-depth investigation in future research.

#### 3.3.5. KEGG Functional Annotation and Enrichment of VOCs

To further investigate the potential biosynthetic pathways associated with differential VOCs in FDG of varying tenderness, the identified VOCs were cross-referenced with the KEGG database to retrieve detailed information on their involved metabolic pathways. Enrichment analysis was subsequently performed to identify pathways, and the annotated results are presented in Figure 9 and Appendix A. A total of eight key VOCs were mapped to four metabolic pathways: butanoate metabolism, monoterpenoid biosynthesis, zeatin biosynthesis, and phenylalanine metabolism. Among them, butanoate metabolism and monoterpenoid biosynthesis were significantly enriched (Figure 9A), indicating their critical roles in the metabolic differentiation of aroma profiles among green teas of different maturity stages. As illustrated in Figure 9B–D and detailed in Appendix A, the butanoate metabolism pathway was associated with 2-propanone, butanal, and methyl acetate; monoterpenoid biosynthesis was linked to nerol, β-pinene, and 1,8-cineole. In contrast, zeatin biosynthesis involved 3-methyl-2-butenal, and phenylalanine metabolism was associated with benzene acetaldehyde.

Pairwise comparison of VOC accumulation patterns revealed that 2-propanone (apple, pear aroma), butanal (chocolate), β-pinene (dry woody), and 3-methyl-2-butenal (sweet, fruity, nutty, almond) were predominantly enriched in FDMJ1G (one bud + one leaf), suggesting enhanced activity of butanoate metabolism and monoterpene biosynthesis at the intermediate maturity stage. In contrast, methyl acetate and benzene acetaldehyde, which contribute sweet, fruity, floral, honey, and cocoa-like aromas, were more highly expressed in FDTC2G (one bud + two leaves). Meanwhile, nerol, a floral monoterpene, was specifically enriched in FDQSG (single bud), supporting its strong floral character.

The KEGG-based enrichment results, when integrated with previous aroma and rOAV analyses, provide a coherent metabolic explanation for the VOC variations observed among the three tenderness grades. Specifically, FDMJ1G was characterized by a notable burst in monoterpenoid biosynthesis and active butanoate metabolism, consistent with its complex and balanced aroma profile. FDQSG was dominated by C6 aldehydes/alcohols (e.g., hexanal, (Z)-3-hexenol), mainly derived from the lipoxygenase (LOX) pathway, contributing to its fresh and grassy aroma [52]. In contrast, FDTC2G showed enrichment in Maillard reaction products, esterification-derived volatiles, and phenylalanine-derived sweet-aromatic compounds such as benzene acetaldehyde, reflecting a maturity-dependent shift toward roasted and sweet-fruity notes.

Taken together, these findings suggest that the differential accumulation of aroma-related VOCs across leaf maturity stages is closely tied to the dynamic modulation of key biosynthetic pathways, particularly butanoate metabolism and monoterpenoid biosynthesis. These pathways may serve as biochemical markers for evaluating and optimizing green tea quality in relation to raw material tenderness.

### 3.4. Identification of Key Flavor Markers of Green Teas of Different Tenderness Based on Six Machine Learning Models

To identify key flavor markers differentiating FDG at distinct maturity stages, data from three complementary analytical platforms—electronic tongue (E-tongue), electronic nose (E-nose), and HS-GC-MS—were comprehensively integrated. The feature set comprised 8 E-tongue sensors, 4 E-nose gas sensors, 31 VOCs with VIP > 1 identified through the PLS-DA model, and 41 aroma-active compounds with rOAV > 1.

These selected features were subjected to classification analysis using six supervised machine learning algorithms: KNN, RF, SVM, GNB, LR, and DT. To enhance model robustness and mitigate overfitting, a 10-fold cross-validation strategy was employed. Classification performance was evaluated using receiver operating characteristic–area under the curve (ROC–AUC), where AUC values between 0.8–0.9 and 0.9–1.0 indicate good and excellent discrimination, respectively.

As illustrated in Figure 10A, five models—KNN, RF, GNB, LR, and DT—achieved perfect classification accuracy (AUC = 1.0), underscoring their high reliability for this task. SVM showed comparatively lower performance, suggesting model-specific limitations for this dataset. Among all models, the RF classifier was prioritized for subsequent analysis due to its well-documented resistance to overfitting and high prediction accuracy in high-dimensional biological datasets [53].

Feature contribution was assessed using Shapley Additive Explanations (SHAP) values derived from the RF model. The top 20 ranked features (Figure 10B) were identified as the most discriminative indicators of tea maturity. Notably, 19 of these features were VOCs detected via HS-GC-IMS, while one corresponded to an E-tongue taste sensor (bitterness). In contrast, no E-nose variables ranked among the top contributors, likely reflecting the broader classification granularity of gas sensors, which may be less sensitive to subtle within-cultivar aroma differences than compound-specific VOCs or taste-related attributes.

Boxplot visualization of the top 20 features (Figure 10C) elucidated their abundance patterns across tea maturity groups. The FDQSG group (single bud) exhibited the greatest number of elevated features (n = 10), including (Z)-3-hexen-1-ol, (E)-2-octenal, propyl isovalerate, n-hexanol-D, octanal-D, 2-ethyl-6-methylpyrazine, n-hexanol-M, nonanal-D, butyl propanoate, and two unidentified VOCs. These compounds conferred a characteristic flavor profile rich in fresh, green, citrusy, herbal, and subtly roasted-potato-like notes, emphasizing the sensory freshness typical of bud-only green tea. In the FDMJ1G group (one bud + one leaf), seven features were enriched, including bitterness (E-tongue), methyl salicylate, 2-methyl-2-pentenal, 2-propanone, n-hexanol-M, 3-methylthiopropanal, and linalool. Collectively, these contributed to a complex and layered aroma composed of floral, fruity, minty, herbal, and vegetable-like elements, reflecting an intermediate degree of leaf maturity and associated biochemical transitions. Conversely, the FDTC2G group (one bud + two leaves) demonstrated elevated levels of only three VOCs—benzene acetaldehyde, 2,5-dimethylfuran, and octanal-M—resulting in a sensory profile characterized by sweet-floral, roasted, cocoa, and mildly herbal notes. These compounds are consistent with enhanced Maillard reaction products and phenylalanine-derived volatiles typically associated with more mature tea leaves.

To assess the discriminative power of individual features, single-variable classification analyses were conducted across all six models, and corresponding ROC curves were generated (Appendix A). Among them, RF and LR models exhibited superior classification performance, indicating their robustness and suitability for feature prioritization. All markers, except for (Z)-3-hexen-1-ol and n-Hexanol-D, achieved individual prediction accuracies above 0.8, confirming their reliability as discriminative features. The slightly lower performance of these two compounds may be attributed to their relatively broad distribution across samples or potential co-elution effects.

In summary, the machine learning-based integration of multimodal sensor data effectively identified robust flavor biomarkers associated with different leaf maturities in FDG. VOCs were the most influential feature group, and the RF and LR models proved most reliable for classification and marker selection. These findings provide a methodological foundation for flavor-targeted quality control and harvest timing optimization in green tea processing.

## 4. Conclusions

This study provides a comprehensive multi-platform analysis of the flavor differences among FDG made from raw materials of three distinct tenderness grades—single bud (FDQSG), one bud plus one leaf (FDMJ1G), and one bud plus two leaves (FDTC2G). Multimodal sensory evaluation using E-tongue, E-nose, and HS-GC-IMS revealed clear differences in taste and volatile compound profiles among the three sample types. FDQSG exhibited higher umami and stronger fresh, green, and herbal notes, with key volatiles such as (Z)-3-hexen-1-ol, (E)-2-octenal, and nonanal reflecting high LOX pathway activity. FDMJ1G presented a complex transitional profile, marked by a peak in linalool (rOAV: 7400), along with floral, fruity, and minty volatiles like methyl salicylate and 2-propanone. FDTC2G, in contrast, was enriched in Maillard-derived and phenylalanine metabolism–related volatiles such as 2-acetyl-1-pyrroline, 2,5-dimethylfuran, and benzene acetaldehyde, contributing to roasted, nutty, and cocoa-like notes. Through KEGG pathway enrichment, four metabolic routes were associated with key volatile formation, notably butanoate metabolism and monoterpenoid biosynthesis, both significantly enriched. Specific markers were also identified for each tenderness grade: Nerol in FDQSG, beta-pinene and 3-methyl-2-butenal in FDMJ1G, and methyl acetate and benzene acetaldehyde in FDTC2G. Machine learning modeling, particularly random forest combined with SHAP analysis, identified 20 top-ranking discriminant markers, with HS-GC-IMS-derived VOCs playing a dominant role. Among them, FDQSG was associated with the highest number of enriched markers (n = 10), reinforcing its sensory profile dominated by freshness and vibrancy. Single-feature ROC–AUC validation confirmed high diagnostic accuracy (≥0.8) for most key indicators, supporting their robustness as maturity-dependent flavor biomarkers.

While this study offers a comprehensive framework for understanding maturity-related flavor shifts in FDG, several limitations remain. Interannual variability in agronomic and climatic factors—such as rainfall patterns, sunlight exposure, and temperature regimes—may influence the biosynthesis and accumulation of both volatile and non-volatile metabolites, thereby affecting flavor expression. These environmental effects should be systematically evaluated in future multi-year studies to consolidate the generalizability of the findings. Moreover, potential sensor cross-sensitivity and matrix effects inherent in E-nose and E-tongue systems may influence the precision of sensor responses and should be carefully considered in future validation efforts. In addition, while the identified volatile compounds with high rOAV values and significant sensor correlations offer valuable insights into the aroma landscape, their direct contributions to perceived flavor require further verification through aroma recombination and omission experiments, as well as human sensory evaluation. To further elucidate the mechanistic basis of taste differences (e.g., umami, bitterness), integration of non-volatile metabolite profiling—such as amino acids and polyphenols—will be essential. Finally, the identity and aroma contribution of unknown VOCs warrant future elucidation via GC–O–MS sniffing techniques. Collectively, these findings provide a molecular basis for understanding the impact of leaf maturity on flavor in FDG. The results offer practical guidance for harvest timing, raw material selection, and grade-specific processing, facilitating more refined quality control and targeted flavor design in the green tea industry.

## Figures and Tables

**Figure 1 foods-14-02861-f001:**
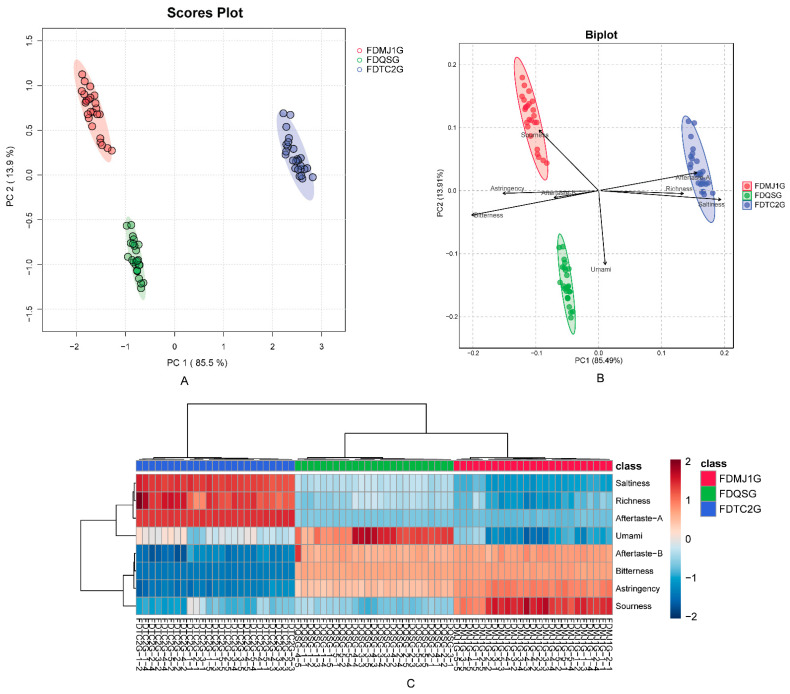
Electronic tongue (e-tongue) analysis of green tea samples with different leaf maturity stages. (**A**) Principal component analysis (PCA) score plot demonstrating sample clustering. (**B**) Biplot integrating sample distribution and variable contributions. (**C**) Heatmap visualizing taste attribute intensities across samples.

**Figure 2 foods-14-02861-f002:**
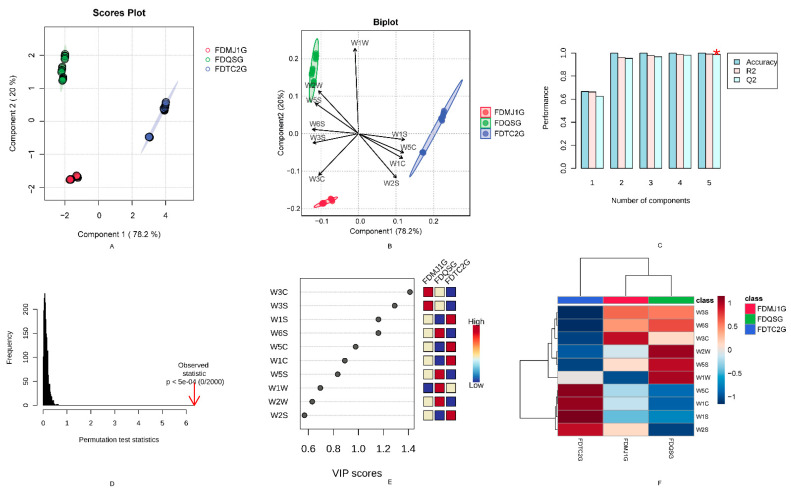
E-nose analysis of taste characteristics in green tea with different leaf maturity stages. (**A**) PLS-DA score plot demonstrating sample clustering based on tenderness grade. (**B**) Biplot with 95% confidence ellipses showing variable contributions to sample discrimination. (**C**) Cross-validation (CV) results indicate model prediction accuracy. Q2 values (predictive ability) with asterisks (*) denote statistical significance (*p* < 0.05), indicating the model outperforms random chance. (**D**) Permutation test statistics (n = 2000) validating model significance (*p* < 0.05). (**E**) Variable importance in projection (VIP) scores plot highlighting key discriminant compounds. (**F**) Heatmap visualization of normalized sensor response patterns across samples (red: high intensity; blue: low intensity).

**Figure 3 foods-14-02861-f003:**
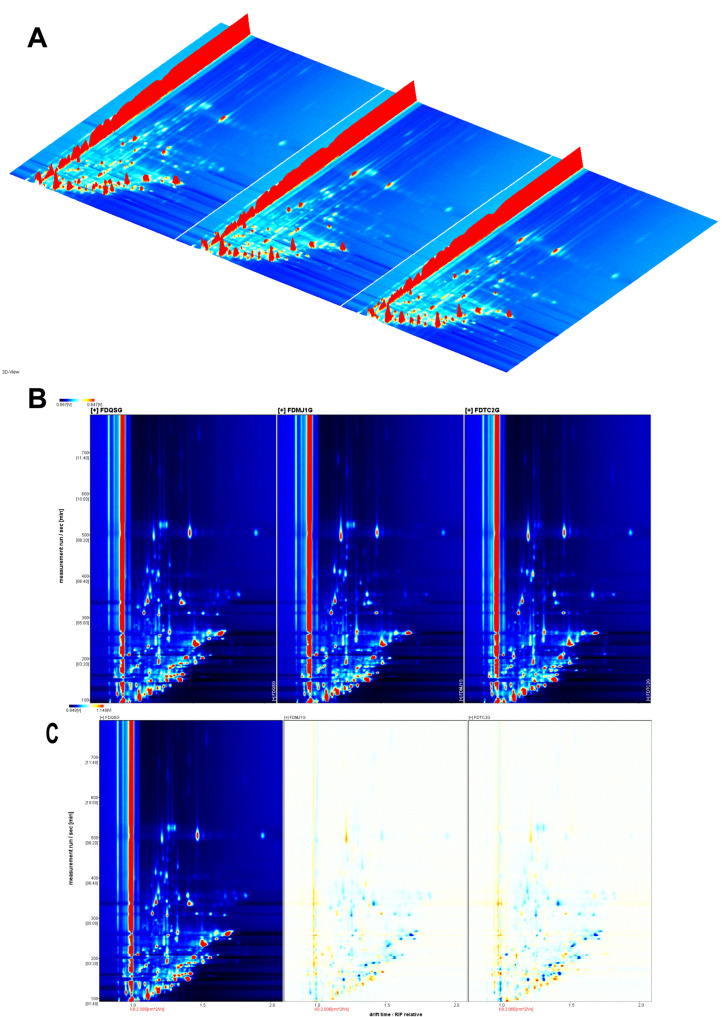
Volatile organic compound profiling of three tenderness-grade green teas using HS-GC-IMS. (**A**) Three-dimensional topographic plots showing ion mobility drift time (*x*-axis), GC retention time (*y*-axis), and normalized signal intensity (*z*-axis, a.u.). (**B**) Two-dimensional topographic plots displaying VOC fingerprint patterns, with RIP (Reaction Ion Peak) normalized to 1.0 (vertical line) and compound signals color-coded by concentration (red: high; blue: low). (**C**) Differential comparison plots generated by spectral subtraction (reference: FDQSG), where red indicates upregulated and blue indicates downregulated VOCs relative to the reference.

**Figure 4 foods-14-02861-f004:**
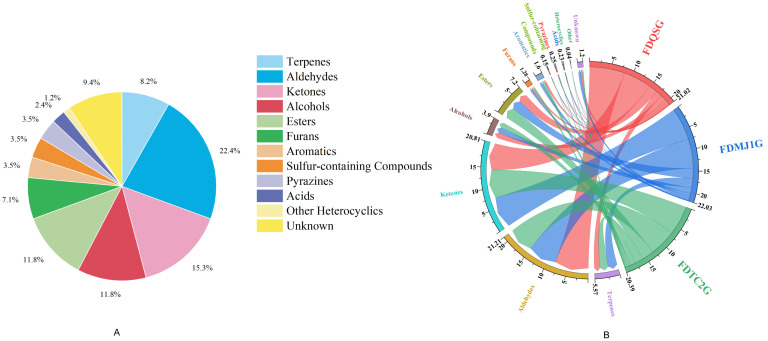
Qualitative and quantitative analysis of volatile organic compounds (VOCs) in green tea samples. (**A**) Proportional distribution of VOC chemical classes (pie chart with percentage annotations). (**B**) Chord diagram illustrating compound-specific correlations between tenderness grades and VOC abundances (band width corresponds to relative concentration).

**Figure 5 foods-14-02861-f005:**
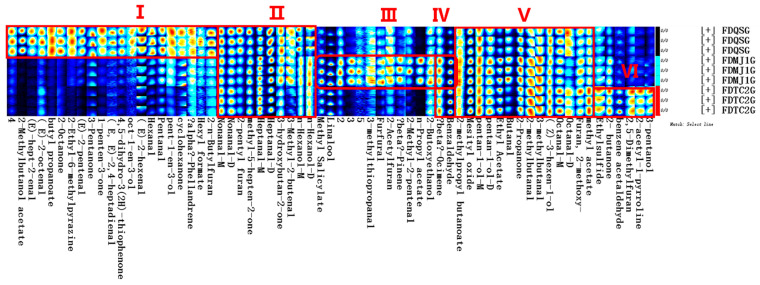
Characteristic VOC fingerprints of FDG across different tenderness grades. Roman numerals indicate clusters of VOCs: I, predominantly enriched in FDQSG; III, predominantly enriched in FDMJ1G; VI, predominantly enriched in FDTC2G; II and V, co-enriched across all FDG grades; IV, predominantly co-enriched in FDMJ1G and FDTC2G.

**Figure 6 foods-14-02861-f006:**
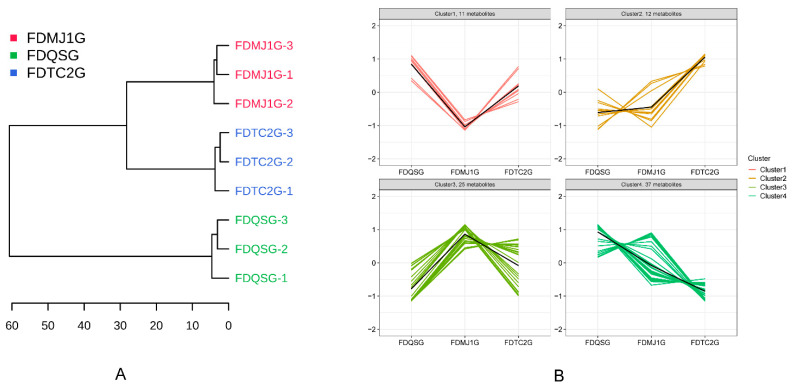
Cluster analysis of VOCs in three tenderness grades of green tea. (**A**) Hierarchical clustering analysis (HCA) dendrogram showing similarity relationships. (**B**) K-means clustering plot demonstrating VOC grouping patterns.

**Figure 7 foods-14-02861-f007:**
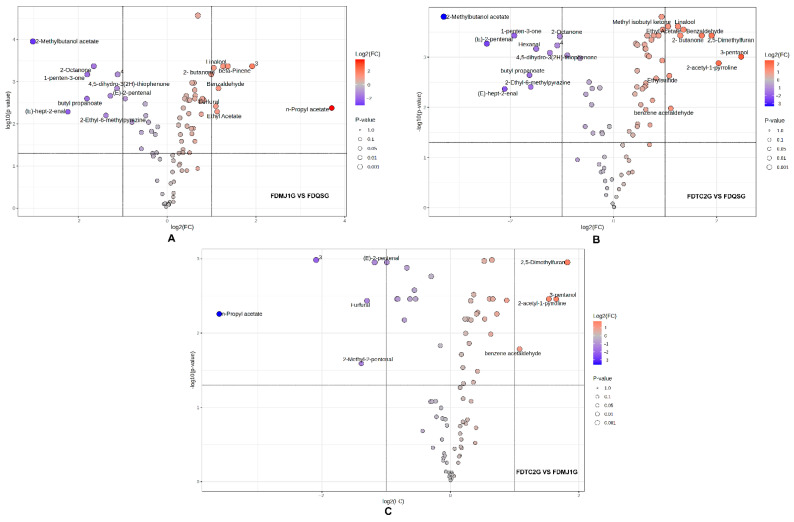
Comparative analysis of VOCs among three tenderness-grade green teas using volcano plots. (**A**) FDMJ1G vs. FDQSG (thresholds: |log2FC| > 1.0, *p* < 0.05). (**B**) FDTC2G vs. FDQSG (thresholds: |log2FC| > 1.0, *p* < 0.05). (**C**) FDTC2G vs. FDMJ1G (thresholds: |log2FC| > 1.0, *p* < 0.05).

**Figure 8 foods-14-02861-f008:**
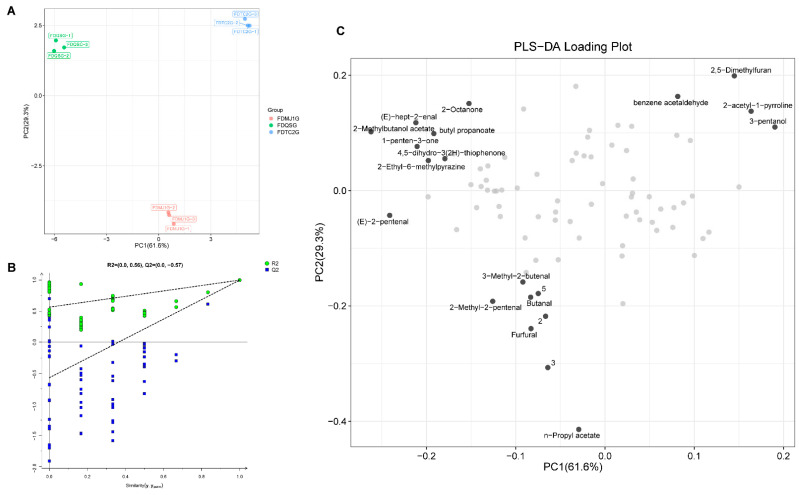
Multivariate analysis of green tea tenderness grades using partial least squares-discriminant analysis (PLS-DA). (**A**) PLS-DA score plot demonstrating group separation. (**B**) Permutation test results (n = 200) validating model robustness. (**C**) Loading plot highlighting highly correlated compounds contributing to group discrimination. Both grey and black circles represent VOCs, with the black circles marking the top 20 VOCs with the highest explanatory contribution to the original variables.

**Figure 9 foods-14-02861-f009:**
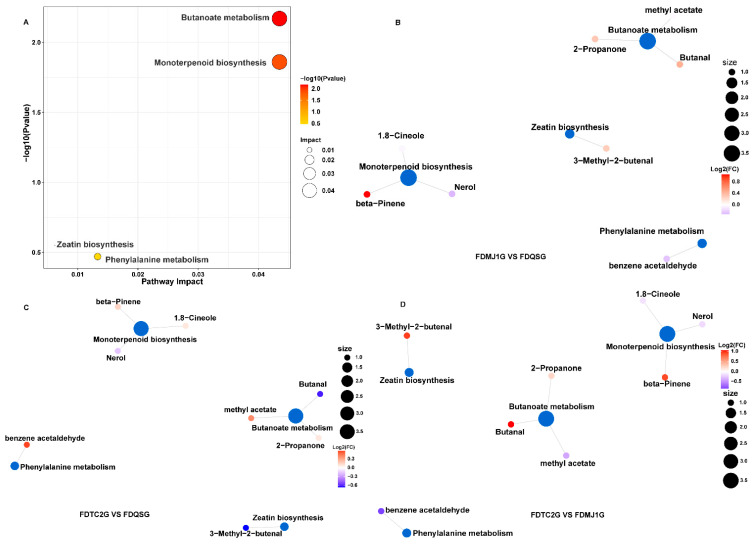
KEGG pathway enrichment analysis of green tea metabolites across leaf maturity stages. (**A**) Scatter plot of enriched pathways. (**B**–**D**) Pathway-metabolite networks (blue: pathways; color gradient: log2FC).

**Figure 10 foods-14-02861-f010:**
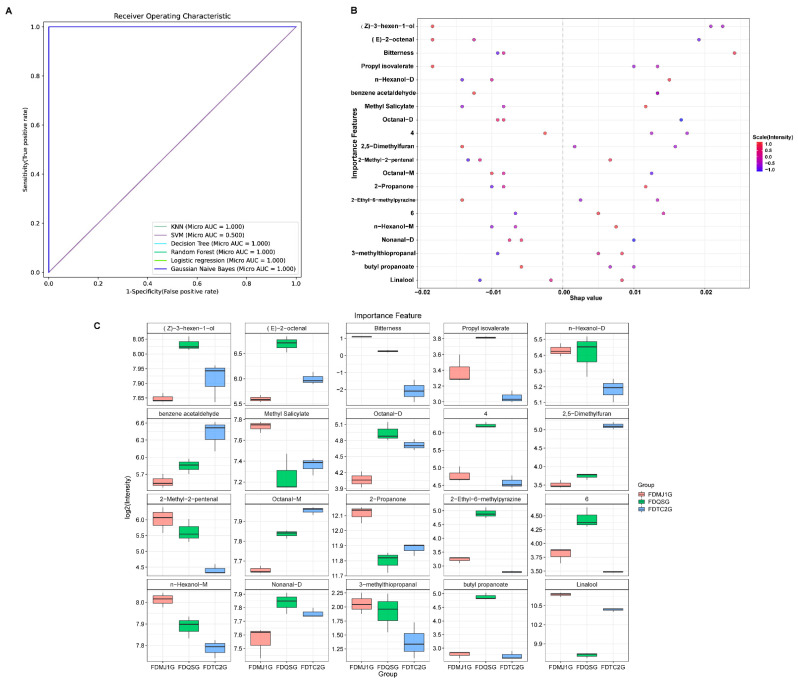
Machine learning-driven identification of flavor markers in FDG of different tenderness grades. (**A**) Receiver operating characteristic (ROC) curves of six algorithms (KNN, RF, SVM, GNB, LR, DT) using all features. (**B**) Bee swarm plot of top 20 discriminant features ranked by SHAP importance. (**C**) Box plots of selected markers’ relative abundance.

**Table 1 foods-14-02861-t001:** Key aroma-active compounds in three tenderness grades of green tea identified using HS-GC-IMS-OAV analysis (rOAV > 1).

NO.	Volatile Compounds	Odor Type	Chemical Classes	OT (μg/L)	rOAV		
					FDQSG	FDMJ1G	FDTC2G
1	Linalool	floral	Terpenes	0.22	3.8 × 10^3^	7.4 × 10^3^	6.3 × 10^3^
2	trans-linalool oxide	floral	Terpenes	60	4.2	3.5	4.0
3	Nonanal-M	aldehydic	Aldehydes	1.1	9.1 × 10^2^	8.1 × 10^2^	8.6 × 10^2^
4	Nonanal-D	aldehydic	Aldehydes	1.1	2.1 × 10^2^	1.7 × 10^2^	2.0 × 10^2^
5	(E)-2-octenal	fatty	Aldehydes	3.0	35	16	21
6	Octanal-M	aldehydic	Aldehydes	0.587	389	343	422
7	Heptanal-M	green	Aldehydes	2.80	135	166	171
8	Heptanal-D	green	Aldehydes	2.80	736	514	423
9	(E)-hept-2-enal	green	Aldehydes	13	5.6	0.95	0.89
10	Octanal-D	aldehydic	Aldehydes	0.587	52.7	28.6	44.8
11	Pentanal	fermented	Aldehydes	12	66	45	25
12	2-methylbutanal	cocoa	Aldehydes	1.0	1.1 × 10^3^	1.3 × 10^3^	1.4 × 10^3^
13	3-methylbutanal	aldehydic	Aldehydes	1.1	5.8 × 10^2^	7.3 × 10^2^	6.9 × 10^2^
14	Butanal	chocolate	Aldehydes	2.0	69	94	45
15	Hexanal	green	Aldehydes	2.4	1.7 × 10^2^	71	41
16	(E)-2-hexenal	green	Aldehydes	88.7	5.96	3.43	2.62
17	methyl-5-hepten-2-one	citrus	Ketones	68	8.5	8.8	7.6
18	2-Octanone	earthy	Ketones	50.2	2.74	0.71	0.92
19	3-hydroxybutan-2-one	buttery	Ketones	14	9.7	9.5	7.0
20	2-Propanone	solvent	Ketones	832	4.28	5.33	4.53
21	1-penten-3-one	spicy	Ketones	23	5.9	1.4	1.1
22	3-Pentanone	ethereal	Ketones	40	3.0	1.9	1.5
23	n-Hexanol-M	herbal	Alcohols	5.6	42.3	46.1	39.4
24	(Z)-3-hexen-1-ol	green	Alcohols	3.9	67	59	62
25	n-Hexanol-D	herbal	Alcohols	5.6	7.6	7.7	6.5
26	1.8-Cineole	herbal	Alcohols	1.10	11.7	11.4	12.5
27	oct-1-en-3-ol	earthy	Alcohols	1.5	33	22	17
28	pentan-1-ol-M	fermented	Alcohols	150.2	2.30	2.40	2.41
29	pentan-1-ol-D	fermented	Alcohols	150.2	1.56	1.66	1.66
30	2-Methylbutanol acetate	fruity	Esters	5.00	25.9	2.58	1.83
31	Propyl isovalerate	fruity	Esters	8.70	1.61	1.21	0.96
32	Ethyl Acetate	ethereal	Esters	5.00	54.5	95.9	92.3
33	methyl acetate	ethereal	Esters	1500	1.080	1.070	1.430
34	2-pentyl furan	fruity	Furans	5.80	19.0	13.8	11.6
35	2-n-Butylfuran	spicy	Furans	5.0	5.8	3.4	3.0
36	Methyl Salicylate	minty	Aromatics	40	3.8	5.3	4.1
37	benzene acetaldehyde	green	Aromatics	6.3	9.2	7.6	14
38	3-methylthiopropanal	vegetable	Sulfur-containing Compounds	0.45	8.5	9.3	5.9
39	Ethylsulfide	coffee, meat	Sulfur-containing Compounds	4.8	3.6	3.7	5.2
40	2-Ethyl-3,5-dimethylpyrazine	nutty	Pyrazines	0.040	1.42 × 10^3^	1.44 × 10^3^	1.35 × 10^3^
41	2-acetyl-1-pyrroline	popcorn	Other Heterocyclics	0.12	57	66	1.6 × 10^2^

## Data Availability

The original contributions presented in this study are included in the article/Appendix A; further inquiries can be directed to the corresponding authors.

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
