# Peer review of "Characterization and Exploration of the Flavor Profiles of Green Teas from Different Leaf Maturity Stages of *Camellia sinensis* cv. Fudingdabai Using E-Nose, E-Tongue, and HS-GC-IMS Combined with Machine Learning"

_foods, 2025, doi:10.3390/foods14162861_

Round 1

Reviewer 1 Report

Comments and Suggestions for Authors

This manuscript investigates how the tenderness level of fresh tea shoots (Camellia sinensis cv. Fudingdabai) influences the sensory and chemical flavor profile of processed green tea. The authors apply a combination of electronic nose (E-nose), electronic tongue (E-tongue), and headspace-gas chromatography-ion mobility spectrometry (HS-GC-IMS), complemented with machine learning models, to characterize volatile and taste-active compounds. The integration of these technologies offers a modern and systematic approach to sensory science.

The title clearly reflects the scope, species, cultivar, and methods.

Abstract:

Include some values in the abstract (e.g., number of volatiles detected and their concentrations)

Introduction:

Indicate properties of green tea and health benefits. 

More detailed explanation of how tenderness physiologically affects flavor precursor metabolism would strengthen the background.

The term “tenderness level” could be better defined (e.g., number of leaves or bud status)

Cite more recent comparative sensory studies on tea maturity

Materials and Methods:

Lack of detailed information on replicates number

Results and Discussion instead Results should be stated:

Increase the dimensions of the Figures

Discuss potential correlations between specific volatiles and sensor responses

Clarify limitations: e.g., sensor cross-sensitivity or matrix effects

Author Response

Author Response to Reviewer Comments

We would like to thank you for their time, thoughtful comments, and valuable suggestions, which have greatly helped improve the quality and clarity of our manuscript. We have carefully addressed each point and revised the manuscript accordingly. Our responses are detailed below.

Reviewer 1:

Comments and Suggestions for Authors

This manuscript investigates how the tenderness level of fresh tea shoots (Camellia sinensis cv. Fudingdabai) influences the sensory and chemical flavor profile of processed green tea. The authors apply a combination of electronic nose (E-nose), electronic tongue (E-tongue), and headspace-gas chromatography-ion mobility spectrometry (HS-GC-IMS), complemented with machine learning models, to characterize volatile and taste-active compounds. The integration of these technologies offers a modern and systematic approach to sensory science.

The title clearly reflects the scope, species, cultivar, and methods.

Abstract:

  1. Comment: Include some values in the abstract (e.g., number of volatiles detected and their concentrations)

Reply:

Thank you for your constructive suggestion. We have revised the abstract to incorporate quantitative details for greater clarity. Specifically, a total of 85 volatile compounds (VOCs) were identified, of which 41 had rOAV > 1. Notably, 2-methylbutanal, 2-ethyl-3,5-dimethylpyrazine, and linalool exhibited ex-tremely high rOAVs (>1000). FDQSG was enriched with LOX (lipoxygenase) -derived fresh, grassy volatiles such as (Z)-3-hexen-1-ol and nonanal. FDMJ1G showed a pronounced accumulation of floral and fruity compounds, especially linalool (rOAV > 7400), while FDTC2G featured Maillard- and phenylalanine-derived volatiles like benzene acetaldehyde and 2,5-dimethylfuran, contributing to roasted and cocoa-like aromas. These data have now been integrated into the abstract to provide a more informative summary of the main findings.

Introduction:

  1. Comment: Indicate properties of green tea and health benefits.

Reply:

We appreciate the suggestion. The introduction section has been revised to include descriptions of the health benefits of green tea, including its antioxidant, anti-inflammatory, neuroprotective, anticancer, cardioprotective, antihyperglycemic, and anti-obesity effects, supported by relevant citations.

  1. Comment: More detailed explanation of how tenderness physiologically affects flavor precursor metabolism would strengthen the background.

Reply:

We have expanded the discussion in the introduction, explaining how tenderness affects the accumulation of flavor precursors such as amino acids and fatty acids, which modulate the production of volatiles during processing. Recent literature has been cited to support this explanation.

  1. Comment: The term “tenderness level” could be better defined (e.g., number of leaves or bud status)

Reply:

We agree and have revised the term “tenderness level” throughout the manuscript as “leaf maturity stages,” specifying each stage as follows: FDQSG (single bud), FDMJ1G (one bud + one leaf), and FDTC2G (one bud + two leaves).

  1. Comment: Cite more recent comparative sensory studies on tea maturity

Reply:

Recent studies focusing on tea leaf maturity and sensory characteristics in green tea, Lithocarpus litseifolius (sweet tea), mulberry leaf tea, black tea, and white tea, have been added and discussed in the introduction to strengthen the background.

Materials and Methods:

  1. Comment: Lack of detailed information on replicates number

Reply:

We have updated the Materials and Methods section to clarify that each analytical technique (E-nose, HS-GC-IMS, E-tongue,) was performed with three or five independent biological replicates per sample group.

  1. Comment: Results and Discussion instead Results should be stated:

Reply: Thank you for pointing this out. We have revised [Results] into [Results and Discussion] to address this point

  1. Comment: Increase the dimensions of the Figures

Reply:

Thank you for your suggestion. We have increased the font sizes of figures for improved readability, with clearer labels and optimized layouts.

  1. Comment: Discuss potential correlations between specific volatiles and sensor responses

Reply:

We have added a subsection in the Results and Discussion highlighting the correlation between specific volatile compounds (e.g., 2-acetyl-1-pyrroline, methyl acetate, C6–C9 aldehydes, ethyl sulfide) and E-nose/E-tongue sensor responses (e.g., bitterness, sourness, richness, saltiness, W1S/W2S/W5C/W1C). This helps connect instrumental data with chemical composition.

  1. Comment: Clarify limitations: e.g., sensor cross-sensitivity or matrix effects

Reply:

Thank you for your suggestion. A paragraph discussing the limitations has been added to the Conclusion. We acknowledge the potential cross-sensitivity and matrix effects of sensors, and the need for further validation through GC-O-MS and aroma recombination/omission studies.

We have carefully revised the manuscript based on the above responses. All changes are highlighted in the revised version. We sincerely thank you and editor for your time and constructive feedback, which greatly improved the manuscript.

Sincerely,

Xiaohui Liu

Reviewer 2 Report

Comments and Suggestions for Authors

The article submitted to the journal, entitled “Characterize and explore the flavour profiles of green teas from different tenderness levels of Camellia sinensis cv. Fudingdabai by E-nose, E-tongue and HS-GC-IMS combined with machine learning” presents a laboratory study aimed at understanding how leaf tenderness influences flavour attributes in green tea, what is crucial for optimizing harvest timing and processing strategies.

The sensory characteristics of green tea leaf samples collected at three distinct stages were subjected to comprehensive analysis. This comprised electronic tongue, electronic nose and GC-MS analysis of volatiles. A sophisticated statistical analysis, incorporating machine learning techniques, was employed to characterise the observed differences and identify the key components.

The manuscript is clearly structured and written in a language understandable to potential readers. The methodology is congruent with the research objectives. The results obtained have been processed using sophisticated statistical methods and are discussed in structured sections. The article contains data and knowledge that will be of use in further research.

Despite the experiment having been conducted on a single exemplary sample, thus limiting the broader applicability of machine learning methods and the potential generalisation of the discovered relationships, the study's findings can contribute significantly to our understanding of the problem, particularly in light of the complexity and sophistication of the methods employed.

I have the following comments on the article:

  1. Ad 2.1 : Were leaf samples taken from different parts of the tea bush and then homogenized? Especially for the sample (1.5g) for the electronic tongue, it is not clear how an average sample was ensured.
  2. I understand that the aim of this work was to characterize sensory properties, sensory active compounds and sensory descriptors using machine analytical techniques. Given the synergistic, masking and antagonistic relationships of the different sensory active substances, it would have been appropriate to complement the experiment with a tasting.
  3. L 219: A link to the website of an external statistical software would be more appropriate in the references.
  4. L238-239: amino acids have not been analysed; a literature citation would be appropriate to support the claim.

Author Response

We would like to thank you for their time, thoughtful comments, and valuable suggestions, which have greatly helped improve the quality and clarity of our manuscript. We have carefully addressed each point and revised the manuscript accordingly. Our responses are detailed below.

I have the following comments on the article:

  1. Comment: Ad 2.1 : Were leaf samples taken from different parts of the tea bush and then homogenized? Especially for the sample (1.5g) for the electronic tongue, it is not clear how an average sample was ensured.

Reply:

Thank you for pointing this out. We have clarified in Section 2.1 that all fresh tea shoots were collected from equivalent positions on the tea bushes to ensure sample consistency. The materials were then thoroughly homogenized before being used for subsequent analyses with the electronic tongue (n = 5), electronic nose (n = 3), and HS-GC-IMS (n = 3).

  1. Comment: I understand that the aim of this work was to characterize sensory properties, sensory active compounds and sensory descriptors using machine analytical techniques. Given the synergistic, masking and antagonistic relationships of the different sensory active substances, it would have been appropriate to complement the experiment with a tasting.

Reply:

We sincerely appreciate this insightful comment. Indeed, human sensory evaluation would offer a valuable complement to the instrumental profiling. However, our primary objective was to establish a rapid, objective, and reproducible analytical framework using advanced flavor detection and computational tools. We fully acknowledge this limitation and have added a statement in the Conclusion section to highlight the need for future validation through aroma recombination and omission tests, as well as formal sensory panel evaluations, to confirm the sensory relevance of the identified markers.

  1. Comment: L 219: A link to the website of an external statistical software would be more appropriate in the references.

Reply:

Thank you for your helpful suggestion. We have updated the reference list to include the official website link for the external statistical software used in this study.

  1. Comment: L238-239: amino acids have not been analysed; a literature citation would be appropriate to support the claim.

Thank you for the valuable reminder. We have now included an appropriate literature citation to support this claim and clarify that amino acid-related insights were derived from published studies.

We have carefully revised the manuscript based on the above responses. All changes are highlighted in the revised version. We sincerely thank you and editor for your time and constructive feedback, which greatly improved the manuscript.

Sincerely,

Xiaohui Liu
